# Trends and Racial/Ethnic Differences in Health Care Spending Stratified by Gender among Adults with Arthritis in the United States 2011–2019

**DOI:** 10.3390/ijerph19159014

**Published:** 2022-07-25

**Authors:** Antoinette L. Spector, Emily Matsen, Leonard E. Egede

**Affiliations:** 1Department of Rehabilitation Sciences and Technology, University of Wisconsin-Milwaukee, Milwaukee, WI 53201, USA; aspector@uwm.edu; 2Center for Advancing Population Science, Medical College of Wisconsin, Milwaukee, WI 53226, USA; ematsen@mcw.edu; 3Department of Medicine, Division of General Internal Medicine, Medical College of Wisconsin, Milwaukee, WI 53226, USA

**Keywords:** healthcare expenditures, arthritis, race/ethnicity

## Abstract

The purpose of this study was to determine if there were racial/ethnic differences and patterns for individual office-based visit expenditures by gender among a nationally representative sample of adults with arthritis. We retrospectively analyzed pooled data from the 2011 to 2019 Medical Expenditure Panel Survey of adults who self-reported an arthritis diagnosis, stratified by gender (men = 13,378; women = 33,261). Our dependent variable was office-based visit expenditures. Our independent variables were survey year (categorized as 2011–2013, 2014–2016, 2017–2019) and race/ethnicity (non-Hispanic White, non-Hispanic Black, Hispanic, non-Hispanic Asian, non-Hispanic other/multiracial). We conducted trends analysis to assess for changes in expenditures over time. We utilized a two-part model to assess differences in office-based expenditures among participants who had any office-based expenditure and then calculated the average marginal effects. The unadjusted office-based visit expenditures increased significantly across the study period for both men and women with arthritis, as well as for some racial and ethnic groups depending on gender. Differing racial and ethnic patterns of expenditures by gender remained after accounting for socio-demographic, healthcare access, and health status factors. Delaying care was an independent driver of higher office-based expenditures for women with arthritis but not men. Our findings reinforce the escalating burden of healthcare costs among U.S. adults with arthritis across genders and certain racial and ethnic groups.

## 1. Introduction

Arthritis is a prevalent chronic health condition and a leading cause of disability in the United States (U.S.), particularly among older adults [1,2]. Osteoarthritis is the most common form of arthritis, followed by rheumatoid arthritis, with classic signs and symptoms of joint pain, stiffness, and swelling, as well as detrimental effects to physical and mental functioning [3,4,5,6]. Between 2016–2018, an estimated 58.5 million U.S. adults were living with arthritis and 44% reported arthritis-attributable activity limitations [2]. It is projected that arthritis prevalence will increase to an estimated 78.4 million by 2040, affecting one in four U.S. adults [2,7,8]. Adults with arthritis spend significantly more annually on healthcare expenditures [9,10,11] and the overall financial impact attributed to arthritis was more than $300 billion in direct and indirect costs in 2013 [12], underscoring the substantial economic burden of arthritis, at both the individual and societal levels.

While arthritis and its negative sequelae affect all population groups, women and racial and ethnic minoritized groups experience a disproportionate burden, including a higher arthritis prevalence and more activity limitations [13,14,15,16]. Evaluating group differences in healthcare expenditures provides another way to draw attention to the potential presence of health disparities, such as barriers to health care access and utilization [17,18]. Adults with arthritis routinely consult with primary care and specialty providers to manage arthritis-related joint pain, stiffness, and functional limitations [19]. Yet, research to date has consistently found that, while women with arthritis are more likely to utilize health services, racial and ethnic minoritized groups are less likely to do so for reasons that are multifactorial in nature, including at the individual, provider, and institutional levels [6,20,21]. Moreover, there is emerging evidence of the nuances that exist when evaluating racial/ethnic differences across types of care, which suggests that non-Hispanic White individuals spend more for outpatient services, whereas individual expenditures for racial and ethnic minoritized groups were typically higher on average for emergency department or inpatient care [17,22]. Less utilization of outpatient services by racial and ethnic minoritized groups is salient because it suggests less access to, and engagement in, more cost-effective, preventive care that could slow down disease progression and functional decline [17,22]. Eliminating disparities in healthcare utilization and clinical outcomes among adults with arthritis remains a key public health priority because it is just, cost-effective, and urgent given the projected increases in the arthritis population and minoritized groups in the U.S. in coming years [20,23,24,25].

Given differences in healthcare utilization patterns by both gender and race/ethnicity, there have been growing calls for the use of intersectional approaches within health disparity research [26,27], including among the arthritis population [28], to develop more effective strategies to mitigate health inequities. When employing an intersectional approach, researchers acknowledge and account for multiple, interdependent, and mutually reinforcing social identities that can produce variation in outcomes within groups through “interlocking systems of privilege and oppression (i.e., racism, sexism, heterosexism, classism) [27] (p. 1267).” Greater attention to intersecting social identities within arthritis research can inform a more tailored approach to clinical and public health interventions, and an equitable allocation of resources [27].

The purpose of this study was to determine if there were racial/ethnic differences and patterns for individual office-based visit expenditures by gender among a nationally representative sample of adults with arthritis. To do this, we first stratified the sample by gender a priori, based on well-established evidence that women and men with arthritis have distinctly different experiences [14,15]. We next assessed within-group racial and ethnic differences and patterns for individual office-based visit expenditures, as well as other correlates of individual expenditures, among men and women with arthritis. We chose to focus on office-based physician visit expenditures for two reasons. Office-based visit expenditures occur in the outpatient setting and can provide a more refined measure to evaluate access to and utilization of more cost-effective and preventive services provided by primary care and specialty providers. Additionally, while there is some evidence of higher utilization of primary care physicians and arthritis-related specialty providers among women with arthritis, and less so among racial and ethnic minoritized groups, there remains limited knowledge on the intersections between them [19,29]. Specific research questions to be addressed were as follows: (1) Are there racial and ethnic differences in office-based visit expenditure trends by gender? (2) Are there different racial and ethnic patterns in office-based visit expenditures by gender after accounting for other relevant factors? (3) What are other correlates of office-based visit expenditures by gender?

## 2. Materials and Methods

### 2.1. Data Source and Study Population

To conduct the current study, we used sequential panels of the Medical Expenditure Panel Survey (MEPS) between 2011 and 2019. MEPS is cosponsored by the Agency for Healthcare Research and Quality (AHRQ) and the National Center for Health Statistics and provides a multitude of data on a nationally representative sample of the U.S. non-institutionalized civilian population and their healthcare experiences over a two-year period [30]. The MEPS Household Component (MEPS-HC) files allow for calculation of direct healthcare expenditures, as well as providing information on several socio-demographic and health characteristics [31]. MEPS-HC data is collected via self-report and then validated and supplemented by data provided from hospitals, medical providers, and pharmacies [31]. We retrospectively analyzed data from participants who were aged 18 years and older and had a self-reported diagnosis of arthritis during the study period (2011–2019) for a total of 19,378 men and 33,261 women with arthritis.

### 2.2. Dependent Variables

The dependent variable was mean office-based visit (OBV) expenditures per person. This included the sum of direct payments from all sources (i.e., individual out-of-pocket expenses, private and public insurance, Worker Compensation, and other miscellaneous sources) for office-based physician visits that were reported by respondents in the MEPS-HC files.

### 2.3. Independent Variables

Survey year. We categorized the study period into three different time periods: 2011–2013, 2014–2016, and 2017–2019. Dividing the sample into three time periods ensured a sufficient sample in each category to assess trends in OBV expenditures during the study period.

Race/Ethnicity. The primary independent variable was race and ethnicity. We categorized the sample into five racial and ethnic groups: non-Hispanic white (NHW), non-Hispanic black (NHB), Hispanic (HSP), non-Hispanic Asian (NHA), and non-Hispanic other (NHO). NHO encompasses all other non-Hispanic racial groups, including American Indian/Alaska Native, Native Hawaiian/Pacific Islander, and multiracial.

### 2.4. Covariates

We accounted for several covariates based on the Andersen and Newman Framework of Healthcare Utilization [32,33]. This framework posits that there are factors that predispose the use of healthcare services, such as your demographic characteristics, and others that enable healthcare utilization, such as insurance status and access to services, as well as factors that create a need for services, such as your health status. For the current study, demographic (predisposing) variables included age, marital status, education, employment status, and region. Enabling factors included poverty status, insurance coverage, and healthcare access. Healthcare access was accounted for by including three dichotomized (yes/no) questions that asked if an individual had a usual source of care, had delayed medical care, or was unable to afford or obtain medical care. Health status (need) variables included total number of comorbidities, whether an individual had any functional limitation (yes/no), and we used the Patient Health Questionnaire to assess depression status using a cutoff score of 3 (i.e., 0–2 = not depressed, 3–6 = depressed) [34].

### 2.5. Statistical Analysis

Three main types of analyses were conducted. First, we calculated means, frequencies, and percentages for study variables stratified by gender. Second, trends analysis was performed to assess total OBV expenditure trends over time. Unadjusted means and 95% confidence intervals (CIs) were calculated overall and by race/ethnicity for each gender. One-way ANOVA was used to compare year categories (2011–2013 vs. 2014–2016 vs. 2017–2019) to test for statistically significant differences in total expenditure over time. Third, a two-part model was utilized to assess differences in OBV expenditures among individuals who had positive office-based spending. The first part was a probit model which estimated the probability of zero OBV expenditure compared to positive OBV expenditure. The second part of the model was a generalized linear model (GLM) with gamma distribution and log link to account for the skewedness of the OBV expenditures variable. The average marginal effects were calculated using post estimation commands using STATA SE 2013 (StataCorp LLP, College Station, TX, USA). Statistical significance was recognized at *p* < 0.05 across the analyses.

## 3. Results

Table 1 shows the unweighted and weighted socio-demographic, healthcare access, and health status characteristics of men and women with arthritis during the study period. Among the weighted sample of men with arthritis (N = 221,869,299), there was a mean age of 61.17 years, and the men were predominantly NHW (77.17%), married (64.64%), employed (53.49%), middle (27.93%) or high-income status (45.17%), privately insured (64.79%), and had a usual source of care (87.93%). In addition, the men had a mean of 2.47 comorbidities, just 32.54% had any functional limitation, the majority were not depressed (78.81%), and few had delayed (6.90%), or were unable to access, medical care (3.90%).

Among the weighted sample of women with arthritis (N = 340,564,368), there was a mean age of 61.86 years and 2.25 comorbidities on average. Similar to men with arthritis, the women were predominantly NHW (72.96%), had a usual source of care (90.02%), were not depressed (78.10%), and few had delayed (8.12%) or were unable to access medical care (5.09%). However, in contrast to men with arthritis, fewer women were married (48.84%), in the high-income category (36.40%), or privately insured (59.59%), and more were employed (61.90%), and had a functional limitation (37.86%).

In unadjusted analyses, mean OBV expenditures increased significantly during the study period for both men and women with arthritis (Table 2). Compared to $2240 in 2011–2013, mean OBV expenditures for men increased to $2499 and $2838 in 2014–2016 and 2017–2019, respectively. Unadjusted mean OBV expenditures for women with arthritis were higher than the men’s expenditures throughout the study period and increased from $2390 in 2011–2013, to $2536 and $3097 in 2014–2016 and 2017–2019, respectively. Patterns of racial and ethnic differences in total expenditures varied among men and women with arthritis. Across men and women with arthritis, NHW and NHB adults had significant increases in OBV expenditures during the study period with no differences noted among NHA men or women. However, while Hispanic women and NHO men had significant increases in their OBV expenditures, there were no differences for Hispanic men or NHO women.

Table 3 shows racial and ethnic differences in expenditures among men and women with any OBV expenditures during the study period when accounting for socio-demographic, healthcare access, and health status factors. While there were no significant differences between NHB and Hispanic men with arthritis, relative to NHW men, NHA and NHO men spent $601 and $605 less, respectively. Overall, men spent $281 more in the 2017–2019 time period compared to 2011–2013. In addition, higher levels of education, living in the West region, greater depression symptoms and comorbidities, having any functional limitation, and a usual source of care were each associated with higher OBV expenditures, while being employed and only publicly insured or uninsured was associated with lower expenditures.

Among women with arthritis who had any OBV expenditure (Table 3), there were significant differences between NHW and NHB women, with NHB women spending $317 less on OBV than NHW women on average. No other racial or ethnic differences were found. Overall, women with arthritis also had higher expenditures in 2017–2019, spending $540 more than in 2011–2013. There were several other independent drivers of OBV expenditures among women with arthritis. Unlike men, being in the middle-income category and delaying medical care was associated with more OBV expenditures among women with arthritis, whereas living in the Midwest or South was associated with lesser expenditures. Similar to men with arthritis, higher levels of education, being in the highest income category, and having more depression symptoms and comorbidities, any functional limitation, and a usual source of care were each associated with higher OBV expenditures among women, while being employed and only publicly insured or uninsured was associated with lower expenditures.

## 4. Discussion

Drawing on data from the Medical Expenditure Panel Survey—a nationally representative sample of the non-institutionalized U.S. population [30]—this study examined trends and differences in OBV expenditures among men and women with arthritis by race and ethnicity between 2011 and 2019. Our analysis yielded three key findings. First, per person OBV expenditures increased for both men and women with arthritis during the study period. Second, racial and ethnic patterns in OBV expenditures differed between men and women with arthritis. Third, healthcare access barriers were independent drivers of OBV expenditures for women with arthritis, but not men. Our findings underscore the importance of nuanced health disparity research among adults with arthritis. While women and racial and ethnic minoritized groups living with arthritis each experience health inequities [2,13,14,15,16], elucidating the intersections between them can guide the development of more effective interventions that are both gender-responsive and culturally-tailored.

### 4.1. Expenditure Trends and Differences by Gender, Race, and Ethnicity

Our findings align with prior research on the escalating economic burden of arthritis in the U.S which is projected to affect 26% of the adult population by 2040 [8,10,21]. These results amplify growing calls for public health and clinical strategies that can address the modifiable risk factors associated with arthritis. In addition, our study found significant racial differences in the average per person expenses of men and women with arthritis that had any OBV expenditure and, to our knowledge, is the first to identify different racial/ethnic patterns by gender, as well as being one of few to distinguish NHA adults with arthritis. Compared to NHW men with arthritis, NHA and NHO men had significantly lower OBV expenditures, whereas OBV expenditures among NHW women were only higher than NHB women with arthritis when accounting for socio-demographic, healthcare access, and health status factors. While earlier findings have shown lower per person medical expenditures for outpatient care for NHB and NHA groups, overall [17,22], our current findings suggest that, in the case of adults with arthritis, lower expenditures for outpatient care among these groups may be driven largely by NHB women and NHA men.

There is mounting evidence that NHB adults with arthritis experience more interpersonal and structural discrimination, and that experiences of discrimination are associated with poorer pain and functional outcomes [16,35,36]. When accounting for a multitude of factors, including healthcare access barriers, differences in OBV expenditures were only present for NHB women. These findings suggest that NHB women with arthritis may be contending with the compounding effects of both gender and racial discrimination [36,37], and that this may limit their use of outpatient services, which is more cost-effective and better able to slow disease progression and functional decline. Future research is warranted to investigate the unique challenges that NHB women with arthritis face in utilizing outpatient care and potential influences on higher utilization of costlier types of care, such as emergency department visits and hospitalizations. In addition, prior research has found significantly higher utilization of complementary and alternative medicine (CAM) among NHB adults with arthritis [38,39]. MEPS does not account for the use of prayer or thermotherapy, for example, which has been shown to be a key part of arthritis self-management among NHB women [38,39,40]. Future research is needed to understand the specific role that CAM plays in arthritis management for NHB women, including it being a potentially more accessible, affordable, or acceptable option compared to the utilization of primary care and specialty providers.

While we offer some potential explanations for differences in OBV expenditures between NHB and NHW women with arthritis, our understanding of racial and ethnic differences in healthcare expenditures remains nascent as it relates to NHA and NHO men with arthritis. While limited, there is evidence that NHA adults have significantly lower healthcare expenditures than NHW adults, overall, with barriers related to language, citizenship, and nativity being key drivers of these differences [17,22]. Moreover, Chen et al. (2013) found that NHA individuals had expenditure patterns that varied by setting type, with lower physician visits and prescription expenditures, but not for hospitals or emergency departments, which is in alignment with our findings of less OBV expenditures among NHA men [22]. The paucity of research on healthcare spending among NHA and NHO adults with arthritis, and differing gender patterns within these groups in healthcare expenditures, is concerning given that Asian Americans are among the fastest growing racial and ethnic minoritized groups and prior research has shown that American Indian/Alaska Native and multiracial adults experience more severe and high-impact pain compared to all other racial and ethnic groups in the U.S. [41]. Public health research and interventions that prioritize these understudied populations, and the gender differences among them, are urgently needed to ensure an equitable distribution of resources to reduce the burden of arthritis both now and in the future.

### 4.2. Other Drivers of Expenditures by Gender

Our results indicated that there are some notable gender differences in the correlates of OBV expenditures among adults with arthritis. Delaying medical care resulted in $462 more in OBV expenditures among women with arthritis but was not an independent driver of expenditures among men. Prior research has shown that women with arthritis are more likely to have higher medical expenditures, overall [21], and specifically related to the utilization of specialty providers [19]. In this case, the higher OBV expenditures specific to women with arthritis who had to delay care may be an indicator that more intensive service provision is required once outpatient care is sought. There was a higher percentage of women with arthritis who were employed and in lower income categories than men during the study period. Healthcare utilization barriers related to time and work flexibility have been found to be more significant for women, regardless of income level, as well as issues with childcare and transportation being especially challenging for women in lower income categories [42]. Future research is warranted to investigate the extent to which women-specific barriers may result in delayed care, which could allow symptoms to worsen and necessitate more intensive and costlier services. Within the clinical setting, the implementation of data collection instruments that can identify the unmet social needs of all adults with arthritis is an evidence-based strategy that has been used in primary care settings [43,44]. Screening for unmet social needs can inform the gathering of relevant resources that can address identified needs, such as barriers to timely healthcare utilization, and reduce health inequities experienced by women with arthritis.

### 4.3. Limitations

The findings from our study should be considered in light of several limitations. As is true for all cross-sectional studies, our results are not able to conclude any causal relationships to healthcare expenditures and observed differences. In addition, the sample used for this analysis included participants who self-reported an arthritis diagnosis, versus participants who had an arthritis-specific diagnosis code. However, the arthritis prevalence estimates in MEPS have largely agreed with other national surveys and suggest that our approach is a valid way to obtain a nationally representative sample of noninstitutionalized U.S. adults with arthritis, particularly in the context of understanding healthcare expenditures and utilization [45]. Given the increasing prevalence of arthritis and activity limitations as people age, it should be noted that the current study’s sample is not representative of institutionalized adults, including people residing in nursing homes. Our study was also limited regarding our categorization of racial and ethnic groups. While we did expand our racial categories to distinguish the NHA population, we acknowledge that there is still significant heterogeneity within this population, as well as within other racial and ethnic minoritized groups. There is an ongoing need to oversample understudied populations in national surveys and for researchers to use intersectional approaches that can shed light on within-group disparities. In addition, although we accounted for several factors related to socio-demographic characteristics, healthcare access, and health status, this list was not exhaustive and there are likely other relevant variables that we could have included in our analysis. Moreover, we were limited to the variables available within the MEPS dataset, so were unable to account for factors related to disease severity, for example, experiences of discrimination, or CAM utilization which are not captured in the MEPS dataset.

## 5. Conclusions

In summary, our findings reinforce the escalating burden of healthcare costs among U.S. adults with arthritis regardless of gender. Our findings also highlight the nuances that exist in healthcare spending among men and women with arthritis, including differing racial patterns by gender, as well as healthcare access barriers as independent drivers of expenditures for women but not men. As projections show the U.S. population will be increasingly affected by arthritis and related activity limitations, it is incumbent upon public health agencies and health systems to prioritize strategies that can optimize the health and well-being of this population in an equitable manner.

## Figures and Tables

**Table 1 ijerph-19-09014-t001:** Sample characteristics of men and women with a self-reported arthritis diagnosis: United States 2011–2019.

	All	Men	Women
	(n = 52,639)	(n = 19,378)	(n = 33,261)
Race/Ethnicity			
NHW	419,693,009 (74.62%)	171,208,656 (77.17%)	248,484,352 (72.96%)
NHB	64,009,730 (11.38%)	21,920,637 (9.88%)	42,089,093 (12.36%)
Hispanic	48,535,036 (8.63%)	17,374,300 (7.83%)	31,160,736 (9.15%)
NH Asian	14,162,016 (2.52%)	5,256,219 (2.37%)	8,905,797 (2.62%)
NH Other or Multiple Races	16,033,875 (2.85%)	6,109,485 (2.75%)	9,924,390 (2.91%)
Age (continuous)	61.59 ± 13.86	61.17 ± 13.46	61.86 ± 14.04
	(N = 562,433,667)	(N = 221,869,299)	(N = 340,564,368)
Marital Status			
Married	309,745,376 (55.07%)	143,407,888 (64.64%)	166,337,488 (48.84%)
Widowed/Divorced/Separated	192,704,101 (34.26%)	53,497,530 (24.11%)	139,206,571 (40.88%)
Never Married	59,984,190 (10.67%)	24,963,881 (11.25%)	35,020,309 (10.28%)
Education			
<High School	70,216,080 (12.48%)	26,581,612 (11.98%)	43,634,468 (12.81%)
High School or GED	219,635,486 (39.05%)	86,131,808 (38.82%)	133,503,678 (39.20%)
College or more	207,137,876 (36.83%)	82,756,875 (37.30%)	124,381,001 (36.52%)
Missing	65,444,225 (11.64%)	26,399,004 (11.90%)	39,045,220 (11.46%)
Employment Status			
Employed	329,481,959 (58.58%)	118,687,204 (53.49%)	210,794,755 (61.90%)
Not employed/will return to work	231,614,437 (41.18%)	102,516,384 (46.21%)	129,098,053 (37.91%)
Missing	1,337,271 (0.24%)	665,711 (0.30%)	671,560 (0.20%)
Region			
Northeast	102,592,927 (18.24%)	39,431,180 (17.77%)	63,161,747 (18.55%)
Midwest	132,339,179 (23.53%)	53,647,347 (24.18%)	78,691,832 (23.11%)
South	215,983,523 (38.4%)	84,100,509 (37.91%)	131,883,014 (38.72%)
West	111,518,038 (19.83%)	44,690,263 (20.14%)	66,827,775 (19.62%)
Poverty Status			
Poor/Negative	73,759,729 (13.11%)	24,143,084 (10.88%)	49,616,645 (14.57%)
Near poor	28,8714,34 (5.13%)	9,971,414 (4.49%)	18,900,020 (5.55%)
Low income	80,983,829 (14.4%)	28,019,402 (12.63%)	52,964,427 (15.55%)
Middle income	154,650,174 (27.5%)	59,526,635 (26.83%)	95,123,539 (27.93%)
High income	224,168,501 (39.86%)	100,208,765 (45.17%)	123,959,737 (36.40%)
Comorbidity Count	2.34 ± 1.72	2.47 ± 1.74	2.25 ± 1.68
	(N = 562,433,667)	(N = 221,869,299)	(N = 340,564,368)
Insurance Coverage			
Any private	346,691,499 (61.64%)	143,757,961 (64.79%)	202,933,538 (59.59%)
Public only	188,988,436 (33.60%)	67,098,833 (30.24%)	121,889,603 (35.79%)
Uninsured	26,753,731 (4.76%)	11,012,504 (4.96%)	15,741,227 (4.62%)
Functional Limitation			
Yes	201,125,122 (35.76%)	72,189,249 (32.54%)	128,935,873 (37.86%)
No	360,229,814 (64.05%)	149,236,798 (67.26%)	210,993,016 (61.95%)
Missing	1,078,731 (0.19%)	443,252 (0.20%)	635,479 (0.19%)
Patient Health Questionnaire			
Depressed	67,308,777 (11.97%)	24,854,328 (11.20%)	42,454,449 (12.47%)
Not depressed	440,827,287 (78.38%)	174,856,579 (78.81%)	265,970,708 (78.10%)
Missing	54,297,603 (9.65%)	22,158,391 (9.99%)	32,139,211 (9.44%)
Usual Source of Care			
Yes	501,670,327 (89.2%)	195,082,176 (87.93%)	306,588,151 (90.02%)
No	54,195,165 (9.64%)	24,088,287 (10.86%)	30,106,878 (8.84%)
Missing	6,568,175 (1.17%)	2,698,836 (1.22%)	3,869,339 (1.14%)
Delayed medical care			
Yes	42,973,222 (7.64%)	15,315,370 (6.90%)	27,657,853 (8.12%)
No	517,629,666 (92.03%)	205,711,234 (92.72%)	311,918,431 (91.59%)
Missing	1,830,778 (0.33%)	842,695 (0.38%)	988,084 (0.29%)
Could not afford or unable to get medical care			
Yes	25,966,092 (4.62%)	8,648,235 (3.90%)	17,317,857 (5.09%)
No	534,709,720 (95.07%)	212,398,449 (95.73%)	322,311,271 (94.64%)
Missing	1,757,854 (0.31%)	822,615 (0.37%)	935,240 (0.27%)

**Table 2 ijerph-19-09014-t002:** Unadjusted mean office-based expenditure overall and by race and ethnicity among men and women with arthritis: United States 2011–2019.

MEN						
	All *** Mean (95% CI)	NHW *** Mean (95% CI)	NHB ** Mean (95% CI)	Hispanic Mean (95% CI)	NHA Mean (95% CI)	NHO * Mean (95% CI)
2011–2013	$2240 (2093, 2387)	$2403 (2231, 2575)	$1650 (1338, 1962)	$1696 (1306, 2806)	$1980 (883, 3078)	$1246 (867, 1625)
2014–2016	$2499 (2285, 2712)	$2618 (2365, 2871)	$2170 (1734, 2605)	$2151 (1478, 2824)	$1484 (977, 1992)	$2212 (1025, 3398)
2017–2019	$2838 (2689, 3018)	$3056 (2830, 3282)	$2331 (1866, 2796)	$1993 (1551, 2435)	$1984 (1225, 2743)	$2056 (1528, 2584)
WOMEN						
	All *** Mean (95% CI)	NHW *** Mean (95% CI)	NHB ** Mean (95% CI)	Hispanic *** Mean (95% CI)	NHA Mean (95% CI)	NHO Mean (95% CI)
2011–2013	$2390 (2233, 2547)	$2510 (2320, 2700)	$1952 (1661, 2243)	$1951 (1585, 2318)	$1781 (1367, 2195)	$3213 (1063, 5363)
2014–2016	$2536 (2364, 2709)	$2695 (2469, 2921)	$2098 (1758, 2438)	$2017 (1781, 2254)	$1810 (1265, 2356)	$2678 (2096, 3260)
2017–2019	$3097 (2926, 3268)	$3237 (3035, 3439)	$2478 (2183, 2774)	$2719 (2215, 3222)	$2355 (1100, 3611)	$4112 (2740, 5484)

NHW non-Hispanic white; NHB non-Hispanic Black; NHA non-Hispanic Asian; NHO non-Hispanic other; CI confidence interval; Statistically significant difference in expenditure over time in full sample or by race/ethnicity indicated by: * *p*-value < 0.05, ** *p*-value < 0.01, *** *p*-value < 0.001.

**Table 3 ijerph-19-09014-t003:** Marginal costs from two-part regression model of office-based visit expenditures among men and women with arthritis: United States 2011–2019.

	All	Men	Women
	Marginal Costs (95% CI)	Marginal Costs (95% CI)	Marginal Costs (95% CI)
Race/Ethnicity			
NHW (REF)	-	-	-
NHB	−284.01 ***	−272.68	−317.06 *
Hispanic	−50.55	−153.64	−17.26
NHA	−315/16	−601.49 ***	−182.59
NHO	−161.73	−604.84 **	135.32
Age			
18–34 (REF)	-	-	-
35–54	22.27	80.28	53.94
55–74	53.75	270.56	−34.90
75+	−19.52	408.58	−254.59
Marital Status			
Married (REF)	-	-	-
No longer married	42.85	−67.80	110.15
Never Married	148.11	25.48	236.85 *
Education			
<High School (REF)	-	-	-
High School or GED	508.20 ***	563.23 ***	456.37 ***
College or more	852.48 ***	750.00 ***	872.13 ***
Region			
Northeast (REF)	-	-	-
Midwest	−189.02 *	−70.52	−259.97 *
South	−243.05 *	−24.10	−382.82 ***
West	23.33	235.30 *	−91.71
Poverty Status			
Poor/Negative (REF)	-	-	-
Near poor	140.75	108.82	173.55
Low income	58.40	−66.97	134.27
Middle income	259.08 ***	168.16	312.46 **
High income	697.49 ***	552.00 **	823.62 ***
Comorbidity Count	245.83 ***	241.64 ***	247.02 ***
Insurance Coverage			
Any private (REF)	-	-	-
Public only	−402.94 ***	−317.96 ***	−460.84 ***
Uninsured	1068.06 ***	−833.16 **	−1215.73 **
PHQ-2			
Not depressed (REF)	-	-	-
Depressed	298.63 ***	367.56 **	249.85 *
Usual Source of Care			
No (REF)	-	-	-
Yes	755.37 ***	683.35 *	742.32 **
Delayed Medical Care			
No (REF)	-	-	-
Yes	405.97 ***	305.05	461.68 **
Could Not Afford or Unable to Get Medical Care			
No (REF)	-	-	-
Yes	−38.66	−129.15	−7.48
Year			
2011–2013 (REF)	-	-	-
2014–2016	46.63	56.27	40.79
2017–2019	436.36 ***	280.91 **	539.91 ***

NHW non-Hispanic white; NHB non-Hispanic Blac; NHA non-Hispanic Asian; NHO non-Hispanic other; CI confidence interval; REF reference group; GED General Education Diploma; PHQ-2 Patient Health Questionnaire-2; Statistically significant difference indicated by: * *p*-value < 0.05, ** *p*-value < 0.01, *** *p*-value < 0.001.

## Data Availability

The data presented in this study are openly available from the Agency for Healthcare Research and Quality at https://meps.ahrq.gov/data_stats/download_data_files_results.jsp?cboDataYear=All&cboDataTypeY=1%2CHousehold+Full+Year+File&buttonYearandDataType=Search&cboPufNumber=All&SearchTitle=Consolidated+Data.

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
