# Peer review of "Trends and Racial/Ethnic Differences in Health Care Spending Stratified by Gender among Adults with Arthritis in the United States 2011–2019"

_ijerph, 2022, doi:10.3390/ijerph19159014_

Round 1

Reviewer 1 Report

I find the topic of the reviewed paper, entitled Trends and Racial/ethnic differences in healthcare expenditure stratified by gender among adults with arthritis in the United States 2011 - 2019, interesting and up to date. However, the Authors should improve the quality of the manuscript.

Below there is a list of my remarks (substance and technical) on the reviewed paper:

·         Please present the same aim of the study in the abstract and the introduction section.

·         The theoretical background presented in the introduction section should be developed.

·         I cannot see the visible contribution of the study. Please formulate it and present it in the introduction section.

·         Please formulate the research hypothesis(es) or research question(s).

·         Please justify the selection of variables for the model (based on the literature review).

·         The research method used in the study is relatively simple but entirely sufficient.

·         I recommend moving the paragraph “limitations” to the end of the discussion section.

Overall assessment

I find the topic of the reviewed paper valuable. However, the Authors should develop the reviewed paper to improve its quality. I recommend making changes (considering the remarks mentioned above) to improve the substance of the manuscript. 

Author Response

-Please present the same aim of the study in the abstract and the introduction section.

We appreciate this feedback. The aim of the study in the abstract now matches the introduction section and reads as follows:

The purpose of this study was to determine if there were racial/ethnic differences and patterns for individual office-based visit expenditures by gender among a nationally representative sample of adults with arthritis.

-The theoretical background presented in the introduction section should be developed.

We agree that the theoretical background for the study could be strengthened. To address this, we have provided additional information in the introduction with respect to taking an intersectional approach to health disparity research. See pg. 2, paragraph 2, lines 68-74.

-I cannot see the visible contribution of the study. Please formulate it and present it in the introduction section.

We agree that the contribution of the study should be more explicit. To address this we have restructured the introduction so that there is a statement of impact at the end of paragraphs 1-3.

P1 (pg. 1, lines 40-43): Adults with arthritis spend significantly more annually on healthcare expenditures [9-11] and the overall financial impact attributed to arthritis was more than $300 billion in direct and indirect costs in 2013 [12]—underscoring the substantial economic burden of arthritis at both the individual and societal level.

P2 (pg. 2, lines 61-64): Eliminating disparities in healthcare utilization and clinical outcomes among adults with arthritis remains a key public health priority because it is just, cost-effective, and urgent given the projected increases in the arthritis population and minoritized groups in the U.S. in coming years [20], [23]–[25].

P3 (pg. 3, lines 72-74): Greater attention to intersecting social identities within arthritis research can inform a more tailored approach to clinical and public health interventions, and an equitable al-location of resources [27].  

-Please formulate the research hypothesis(es) or research question(s).

We appreciate this feedback and have now added the research questions at the end of the introduction. They can be found on pg. 2, paragraph 3, lines 88-92 and are shown below:

Specific research questions to be addressed are as follows: (1) Are there racial and ethnic differences in office-based visit expenditure trends by gender? (2) Are there different racial and ethnic patterns in office-based visit expenditures by gender after accounting for other relevant factors? (3) What are other correlates of office-based visit expenditures by gender?

-Please justify the selection of variables for the model (based on the literature review).

We appreciate this feedback and have added information within the Materials and Methods to justify the selection of covariates within the model. Our justification for the selection of covariates can be found on pg. 3, lines 128-132.

Regarding the dependent and independent variables, we provide justification in the introduction on pg. 2, lines 77-88. Here we describe how we stratified the sample based on existing evidence of gender differences and have now added that we were assessing for within-group differences for each gender to explore intersections between gender and race/ethnicity based on limited prior research in this area.

-The research method used in the study is relatively simple but entirely sufficient.

We appreciate that the reviewer found our methodological approach to be appropriate for the study design.

-I recommend moving the paragraph “limitations” to the end of the discussion section.

We agree with this feedback and have now moved the “limitations” paragraph to the end of the discussion section.

Reviewer 2 Report

General comments

The authors present and interesting and informative study of the gender and racial/ethnic patterns in health care expenditures over time among patients with arthritis. Given both the increasing prevalence of arthritis and the increasing health care spending in the United States, this is an important and timely topic. The methods are sound the results are clearly presented. I only have a few minor comments.

Introduction

It is unclear in the second paragraph of the Introduction if the amount spent on health care (lines 54 – 56) are overall or by the patient. The implications of the statement depend greatly on this, so it should be more clearly specified.

Although the authors discuss in the limitation section the concern that medical expenditures are not necessarily a reflection of appropriate use of care, this is not set up in the Introduction. In fact, it is unclear throughout the paper if greater medical expenditures are considered “good” or “bad” outcomes. A more detailed consideration of what medical expenditures actually represent would help the reader when later interpreting the results.

Results

On page 4, when discussing the results of Table 1, the authors list privately insured as being both similar between men and women (line 145) and different between men and women (line 148-149). The authors should clarify which one it is.

Discussion

More time should be devoted to the policy and practice implications of the findings. The authors repeat a few times the importance of understanding the nuance and intersectionality in what drives medical expenditures for this group, but it is mostly unclear what the authors recommend doing with this information.

In the discussion of the limitation of only having access to the variables included in the MEPS dataset, the authors should specifically call out the lack of information on previous experiences of discrimination within the health care system as this is a key driver of racial/ethnic disparities in care.

Author Response

Introduction

It is unclear in the second paragraph of the Introduction if the amount spent on health care (lines 54 – 56) are overall or by the patient. The implications of the statement depend greatly on this, so it should be more clearly specified.

We appreciate the feedback and have clarified that the racial/ethnic differences in spending that we reference are for expenditures. This clarification can be found on pg. 2, lines 54-58.

Although the authors discuss in the limitation section the concern that medical expenditures are not necessarily a reflection of appropriate use of care, this is not set up in the Introduction. In fact, it is unclear throughout the paper if greater medical expenditures are considered “good” or “bad” outcomes. A more detailed consideration of what medical expenditures actually represent would help the reader when later interpreting the results.

The reviewer makes a good point. We have now clarified whether greater expenditures are “good” vs. “bad”. Based on our focus on office-based visit expenditures, we now make a clearer argument that groups who have higher medical expenditures is an indicator of more engagement with care that is cost-effective and better able to slow disease progression and functional decline. We present prior evidence of existing racial/ethnic differences wherein racial/ethnic minoritized groups had greater utilization of emergency department and inpatient care, which is costlier and less effective for disease management, to support this argument. We believe that providing this background information, and references to other studies who make a similar argument, will provide readers with better context when later interpreting results. This clarification can be found on pg. 2, lines 54-61.

In addition, we have removed the statement on medical expenditures not necessarily being a reflection of appropriate use of care from the limitations section. With the addition of more detailed background information to clarify what greater or lesser expenditures represent, we no longer believe this statement is needed within the limitations section and may actually create confusion if we kept it as a limitation.

Results

On page 4, when discussing the results of Table 1, the authors list privately insured as being both similar between men and women (line 145) and different between men and women (line 148-149). The authors should clarify which one it is.

We appreciate this feedback. For clarity, we have now removed the reference to private insurance being similar between men and women and have left the reference to women having a lower percentage who are privately insured (line 170).

Discussion

More time should be devoted to the policy and practice implications of the findings. The authors repeat a few times the importance of understanding the nuance and intersectionality in what drives medical expenditures for this group, but it is mostly unclear what the authors recommend doing with this information.

We agree that there should be a greater emphasis on policy and practice implications. We do mention on pg. 9, lines 228-232 that intersectional approaches can inform the development of more effective interventions that consider multiple identities (i.e., gender and race/ethnicity). In addition, we now draw attention to a need for public health research and interventions that target health disparities affecting Asian Americans, AI/AN, and multiracial adults with arthritis to ensure an equitable distribution of resources to reduce their arthritis burden (see pg. 10, lines 281-284). We also expand on our discussion regarding screening for social needs within the clinical setting to reduce health inequities experienced by women with arthritis (see pg. 10, lines 300-305).

In the discussion of the limitation of only having access to the variables included in the MEPS dataset, the authors should specifically call out the lack of information on previous experiences of discrimination within the health care system as this is a key driver of racial/ethnic disparities in care.

We agree with this point and have added a discussion about discrimination missing from the MEPS dataset in the limitations section (see pg. 11, lines 326-329).

Reviewer 3 Report

The research topic is of scientific and social interest.

There is cohesion in the article, between the section on Theoretical Framework and

the subsequent sections, which describe the study and draw conclusions, have a

correct, well-structured and cohesive design. 

In general, the article is correct and I consider that the topic is in line with the journal’s

research objectives.

INTRODUCTION:

The study objective is well defined and identified in both the abstract and the

introduction.

The subject under investigation is of growing scientific and social interest. The

investigation is current.

MATERIALS, METHODS and RESULTS:

The statistical treatment is correct. The data is well structured.

DISCUSSION and CONCLUSION:

The conclusions are well drawn and interesting.

The discussion is correct.

I would like to see some more conclusions, regarding action, public policies and their possible implementation.

Author Response

I would like to see some more conclusions, regarding action, public policies and their possible implementation.

We agree with this point and have made some changes as described above. We highlight the need for a public health policy and research agenda that prioritizes understudied populations to ensure equitable distribution of resources as these groups make up a larger share of the population and are already disproportionately affected by arthritis. We also expand on our discussion about the implementation of screening for social needs to identify barriers to healthcare services, particularly care that is more cost-effective and better able to prevent disease progression and functional decline. We highlight how this could specifically address gender inequities.

Round 2

Reviewer 1 Report

I am glad that the Authors of the study took into account the vast majority of my remarks. I accept the current version of the manuscript.